evolution, genetics, microbiology

capsule serotype, evolution, biotic stress, diversifying selection

**Author for correspondence:**
Olaya Rendueles
e-mail: olaya.rendueles-garcia@pasteur.fr

# Nutrient conditions are primary drivers of bacterial capsule maintenance in *Klebsiella*

Amandine Buffet, Eduardo P. C. Rocha and Olaya Rendueles

Microbial Evolutionary Genomics, Institut Pasteur, CNRS, UMR3525, Paris 75015, France

AB, 0000-0001-7340-9075; EPCR, 0000-0001-7704-822X; OR, 0000-0002-6648-1594

The fitness cost associated with the production of bacterial capsules is considered to be offset by the protection provided by these extracellular structures against biotic aggressions or abiotic stress. However, it is unknown if the capsule contributes to fitness in the absence of these. Here, we explored conditions favouring the maintenance of the capsule in *Klebsiella pneumoniae,* where the capsule is known to be a major virulence factor. Using short-term experimental evolution on different *Klebsiella* strains, we showed that small environmental variations have a strong impact on the maintenance of the capsule. Capsule inactivation is frequent in nutrient-rich, but scarce in nutrient-poor media. Competitions between wild-type and capsule mutants in nine different strains confirmed that the capsule is costly in nutrient-rich media. Surprisingly, these results also showed that the presence of a capsule provides a clear fitness advantage in nutrient-poor conditions by increasing both growth rates and population yields. The comparative analyses of the wild-type and capsule mutants reveal complex interactions between the environment, genetic background and serotype even in relation to traits known to be relevant during pathogenesis. In conclusion, our data suggest there are novel roles for bacterial capsules yet to be discovered and further supports the notion that the capsule's role in virulence may be a by-product of its contribution to bacterial adaptation outside the host.

## 1. Introduction

Most environments in which bacteria thrive are complex and can be temporally or spatially heterogeneous, both in their abiotic (pH, nutrients, chemicals…) or biotic compositions (niche invasions, extinction events). This may limit the ability of bacteria to efficiently adapt and ultimately survive, and thus represents an evolutionary challenge [1,2]. To withstand unfavourable or changing environments, bacteria can produce a thick extracellular layer—the capsule—which constitutes the first barrier between the cell and its environment. Capsules are encoded in half of the sequenced bacterial genomes as well as in some archaea and yeast [3]. Whereas capsules have been associated with resistance to abiotic stresses such as UV light and desiccation [4,5], they were mostly studied through the prism of their protective role against biotic stresses, including resistance to grazing protozoa [6], antibiotics [7,8], antimicrobial peptides [9,10] and host immune defences, such as human serum and phagocytes [11,12].

The majority of capsules belong to the so-called Group I or Wzx/Wzy-dependent capsules [3]. These capsules are high molecular weight polysaccharides composed of repeated oligosaccharide units. In bacteria, the presence of different sugar-specific genes in the capsule operon results in different oligosaccharide combinations and residue modifications, which leads to different serotypes. Serotype diversity, even within-species, can be very large and has been studied already in several facultative pathogens [13,14]. In *Klebsiella pneumoniae,* there are at least 77 serologically defined capsule serotypes

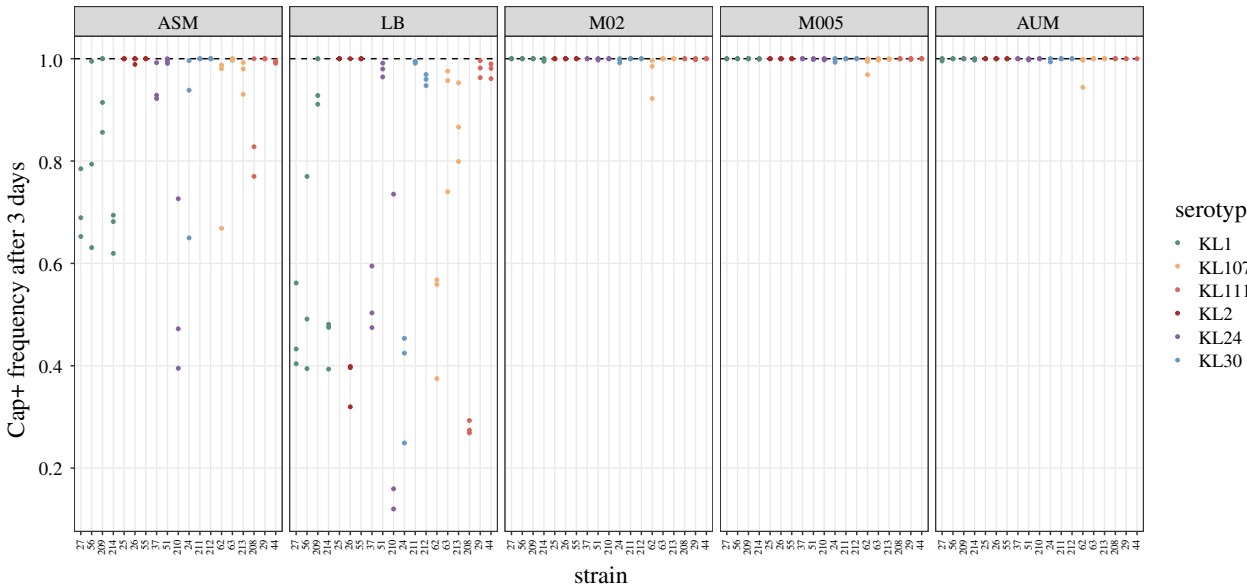

**Figure 1.** Frequency of capsulated colonies across strains and growth media after 3 days of evolution. Each point represents an independently evolved population. Strain KL30.no. 212 was unable to grow in M005. (Online version in colour.)

[15,16], and more than 130 serotypes identified through comparative genomics [17,18]. This serotype diversity results from the faster evolution of capsular loci compared to that of the rest of the genome owing to high rates of homologous recombination [19,20], and horizontal gene transfer [21,22]. As a result, the same capsular serotype can be found in multiple lineages of *K. pneumoniae* and a monophyletic clade can include distinct serotypes [20].

Many studies have assessed the role of the capsule in enhancing virulence. However, the costs and benefits of the capsule when bacteria are not under biotic stress has not been thoroughly addressed. Further, it is also currently unknown whether all capsules (or all serotypes) provide the same advantages and whether their function is evolutionarily conserved. Here, we investigated these questions using as a model the *K. pneumoniae* species complex, hereafter referred to as *Kpn* SC, which encompasses several species, including *Klebsiella variicola. Klebsiella* spp are ubiquitous free-living and host-associated bacteria able to colonize a large range of environments [23,24]. Most isolated *Klebsiella* have a Wzx/Wzy-dependent capsule, regarded as the major virulence factor involved in the bacterial ability to cause both community-acquired and hospital-acquired infections in humans [25,26]. To test the maintenance of capsule production in several growth conditions, including host-related environments, we selected 19 strains representative of the genetic diversity of *Kpn* SC [23]. We also generated a panel of nine capsule mutants to address the precise fitness effects and functions of the capsule in a range of conditions. The integration of these results revealed that the *Kpn* SC capsule can be selected in the absence of biotic stresses, suggesting that its contribution to bacterial adaptation to non-biotic stress and nutrient conditions outside the host may be an important driver of its evolution and maintenance.

## 2. Results

### (a) *Klebsiella* capsule is maintained in nutrient-poor environments

To understand the fitness cost of a capsule in the absence of biotic pressures, we serially transferred 19 different *Kpn* SC strains belonging to six different serotypes (with at least three strains per serotype, electronic supplementary material, table S1), into fresh media every 24 h for 3 days. We then measured the natural emergence of non-capsulated mutants (Cap−) after *ca* 20 generations in five different growth media, including three laboratory media with decreasing levels of nutrients: rich lysogeny broth (LB) medium, minimal medium supplemented with 0.2% of glucose (M02), and minimal medium supplemented with 0.05% of glucose (M005). We also tested artificial sputum medium (ASM) [27] and artificial urine medium (AUM) [28] mimicking host-related nutritional conditions. We first verified by capsule quantification that all tested strains were able to produce a capsule in the different growth conditions (electronic supplementary material, figure S1). Although each growth media had different carrying capacities, i.e. the maximum population size an environment can sustain, all cultures reached bacterial saturation in late stationary phase, ensuring that the different populations underwent a similar number of generations across media.

We observed that Cap− emerged after three days in 111 of the 290 independently evolved populations (38%) (figure 1). Most ASM (74%) and LB (89%) lineages showed capsule mutants, accounting for 85 of the 111 populations with Cap−. A subset of the latter was verified by capsule quantification (electronic supplementary material, figure S2), and whole genome sequencing revealed mutations in the capsule operon, namely in the initial glycosyltransferase, *wcaJ* [29]. Our results also show that capsule inactivation in nutrient-rich environments is not homogeneous across strains nor serotypes, suggesting that the costs differ across strains. We hypothesized that these differences could be owing to the amount of capsule produced by each strain in each environment, but we observed no correlation between the amount of capsule produced by each strain and the frequency of Cap− emergence ($p > 0.05$, Spearman's rank correlation). Using a generalized linear model, we also tested the relevance of serotype and growth media on the emergence of Cap− ($R^2 = 0.39$) and showed that the latter was both dependent on the growth media ($p < 0.001$) and on the serotype ($p < 0.02$).

More importantly, Cap− were very rarely observed in the other three low-nutrient growth media (M02, M005 and

Proc. R. Soc. B 288: 20202876

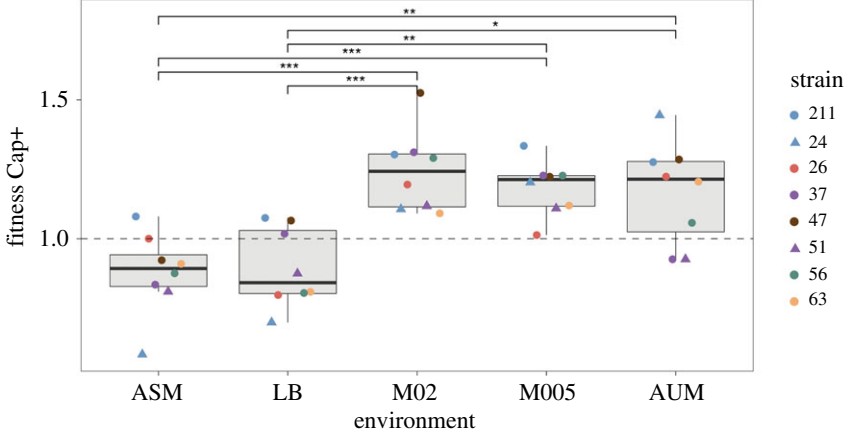

**Figure 2.** Competition between capsule mutants and their associated wild-type in different growth media. Each point represents the average of at least three independent replicates. Individual error bars for each strain are not displayed for clarity purposes. Colours represent different serotypes. Different shapes (dot and triangle) are used to distinguish between strains with the same serotype. Results for the strains with no compensatory mutations are presented in the electronic supplementary material, figure S5. *$p < 0.05$; **$p < 0.01$; ***$p < 0.001$, Tukey post hoc and correction for multiple tests. (Online version in colour.)

AUM), and mostly concerned one strain (KL107.no. 62). Further, we tested whether the capsule could be maintained for periods much longer than the three days of the experiment. Indeed, KL2.no. 26 evolving in M02 maintained its capsule at least for 30 days (electronic supplementary material, figure S3). These results imply that capsule production is deleterious under rich media and selected for in poor media, a phenomenon depending to a lesser extent on the capsule serotype and suggesting that capsule may have other roles than those commonly evoked.

## (b) *Klebsiella pneumoniae* species complex capsule provides a fitness advantage in nutrient-poor environments

To assess the costs of capsule expression across serotypes and growth media, we generated isogenic Cap− mutants for several strains from different serotypes by in-frame and markerless deletions of *wza*. *Wza* is a core gene of the capsule operon and encodes an outer membrane exporter. From the 15 genetically amenable strains we attempted to mutagenize, we obtained clean *wza* deletion mutants for five strains, from several serotypes, whereas four other strains carried off-target compensatory mutations (electronic supplementary material, table S2). Microscopic observations and uronic acid quantifications confirmed that all nine Δ*wza* mutants lack a capsule (electronic supplementary material, figure S4). We performed direct competition experiments between the parental strains and their corresponding Δ*wza* mutant. The strains were initially mixed in a 1 : 1 ratio and allowed to compete for 24 h in the five aforementioned growth media. Competition experiments showed a significant effect of the environment in the fitness of the capsulated (Cap+) strains (figure 2; Kruskal–Wallis, $p = 0.0046$, $n = 8$). In this analysis, we excluded strain KL2.no. 55 owing to a growth deficiency of the Δ*wza* mutant. Overall, we observed a marginal decrease of Cap+ genotypes in nutrient-rich media ($p = 0.07$ and $p = 0.05$ in LB and ASM, respectively, one-sided *t*-test, difference of 1). However, lower fitness in these growth media was significant in 5 out of 8 strains in LB and 4 out of 8 in ASM strains, indicative of variation in costs across strains.

By contrast, Cap+ strains have a strong fitness advantage in low-nutrient media M02, M05 and AUM ($p < 0.05$, $n = 8$). The average fitness advantage provided by the capsule in nutrient-poor media was *ca* 20%. The same trend is observed when only the five clean mutants are taken into account (electronic supplementary material, figure S5A). Individually, most strains had a significant fitness advantage in nutrient-poor environments, with the exception of the two KL24. In these two strains, the capsule significantly imposed a fitness cost in AUM, suggesting a particular negative interaction between the serotype KL24 and AUM (figure 2).

To confirm that the amount of nutrients in the media determined the fitness of Cap+ strains, we competed a subset of strains against their respective Cap− in diluted LB (1 : 10). The chosen strains (KL1.no. 56, KL2.no. 26 and KL30.no. 24) represent three different serotypes and their capsules are significantly costly in LB (figure 2). We observed a fitness increase of Cap+ strains of approximately 23% in the diluted rich media (Wilcoxon rank-sum test, $p < 0.01$ for KL2.no. 26 and KL30.no. 24). Similar qualitative results were obtained in diluted ASM (Wilcoxon rank-sum test, for strain KL30.no. 24, $p < 0.01$). Finally, to control for potential fitness side-effects of *wza* deletion owing to the accumulation of capsule polysaccharide in the periplasmic space [30], we constructed Δ*wcaJ* mutants, a non-core enzyme which is the first glycosyltransferase of the capsule synthesis pathway. We performed the competition experiments in the abovementioned three strains and showed a strong fitness disadvantage of the capsule in rich media. The opposite is true in nutrient-poor media, in which the capsule provides a significant advantage (electronic supplementary material, figure S5B), Kruskal–Wallis, $p = 0.005$. Taken together, our results therefore confirm that the presence of a capsule is disadvantageous in rich media and provides a fitness advantage in nutrient-poor media, albeit differences across strains.

## (c) Capsule increases growth rate and population yield in low-nutrient conditions

To further characterize how the capsule provides a fitness advantage in nutrient-poor media, we grew all strains in microtiter plates and evaluated three different growth parameters: minimum generation time, maximum yield and the

Proc. R. Soc. B 288: 20202876

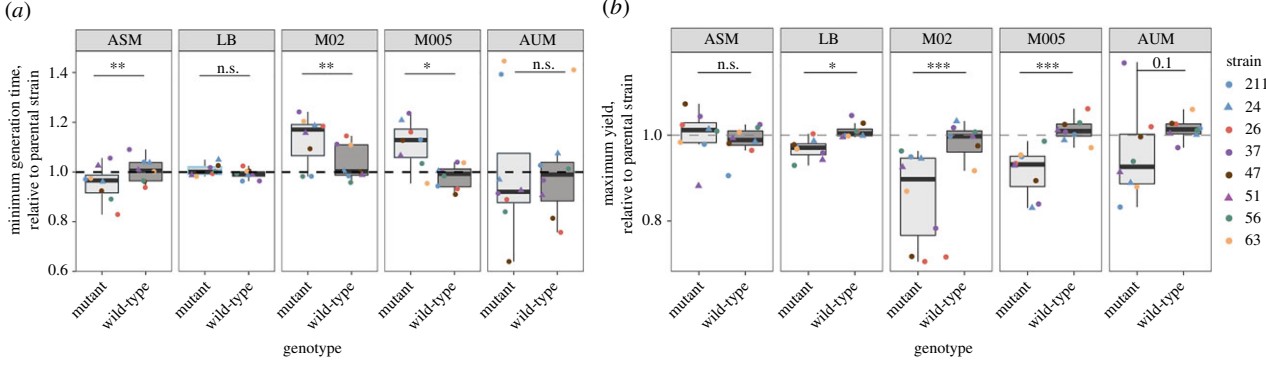

**Figure 3.** Growth dynamics in different growth media. (*a*) Minimum generation time of the mutant and the wild-type strains relative to the parental strain in each media. (*b*) Maximum yield of the mutant and the wild-type strains relative to the parental strain in each media. Statistics were calculated using a paired *t*-test. *$p < 0.05$; **$p < 0.01$; ***$p < 0.001$. (Online version in colour.)

area under the curve (AUC), which takes into account the initial population size, growth rate and carrying capacity of the environment. We also included in our experiments a wild-type strain subjected to the same double-recombination event as the mutant but resulting in a wild-type allele. As expected, we observed significant differences in growth rate across strains and environments (generalized linear model, $p < 0.001$ for both, $R^2 = 0.60$). We show that, in ASM, the Cap+ strains have slower growth rate and achieve lower absorbances (ODs) (figure 3*a* and electronic supplementary material, figure S6A), which is a proxy for population yield as control experiments showed a strong correlation between absorbance and cell numbers (see Methods). This is not the case in LB, in which Cap+ strains reached higher yields compared to ASM. In nutrient-poor media, we observe the opposite to ASM, namely, that the growth rate of Cap+ strains is higher, and the populations reach higher yields (figure 3*b* and electronic supplementary material, figure S6B). Furthermore, direct plating of cells after 16 h of growth in microtiter plates also revealed larger population sizes in Cap+ strains than in Cap− in nutrient-poor media. These observations are qualitatively similar to those observed using AUC calculations, notably in M02 and M005, in which AUC is significantly higher in Cap+ strains compared to Cap− strains (electronic supplementary material, figure S6C), despite large differences across strains. Qualitatively similar results were obtained using the three Δ*wcaJ* capsular mutants, namely, a higher yield and AUC of Cap+ strains in nutrient-poor environments but not in ASM or LB. Taken together, these results suggest that differences in growth rate and yield between Cap+ and Cap− strains in nutrient-poor media could explain the fitness advantages during competition.

### (d) Growth rate and yield increases in capsulated bacteria do not rely on increased survival nor reduced death rate

To understand how the capsule could increase *Kpn* SC growth rate and yield in nutrient-poor environments, we first tested whether capsule production could increase cell survival during starvation, as shown in *Streptococcus pneumoniae* [31]. As expected, mortality rates of *Klebsiella* were very low compared to other Enterobacteria [32], but we observed no significant effect of the capsule after 30 or 60 days (electronic supplementary material, figure S7A). We then tested

if Cap− strains exhibit higher death rates in our experimental conditions. The use of Live-Dead stain and determination of the number of dead cells after 16 h of growth did not show any difference between wild-type and Cap− strains (two-way ANOVA, $p = 0.47$: electronic supplementary material, figure S7B).

We then hypothesized that capsulated bacteria could use their capsule as a nutrient source in nutrient-poor growth media. To test this, we used the Δ*wcaJ* mutants, to avoid capsule build up in the periplasm that could interfere with our results. We allowed the mutant strains to grow for 24 and 48 h in minimal media with no source of carbon, in the presence of purified capsule added exogenously and of uronic acid used as a standard to quantify capsule production. We did not observe significant differences between the treatments where the capsule was added exogenously and the respective controls (Kruskal–Wallis, $p > 0.05$, electronic supplementary material, figure S7C). Our results suggested that the advantage provided by the capsule is not owing to an increased survival during starvation, reduced death rate nor on the ability of the cell to consume the capsule. Thus far, the mechanisms allowing an increased growth rate of Cap+ strains remain unknown.

### (e) The roles of the capsule during pathogenesis are not conserved across strains

Our previous results show that the costs of the capsule are different across strains and serotypes. We thus sought to further understand if the roles of the capsule also differed or were largely conserved in relevant traits affecting pathogenesis. We first addressed the ability of Cap− to resist to human serum. Six Cap− were significantly more sensitive to human serum than their Cap+ counterpart, whereas the other three strains exhibited no differences (figure 4*a*). Neither capsule serotype nor O-antigen serotype (known to provide resistance against human serum [33]) could explain the resistance profiles. Our results strongly suggest that the effect of the capsule on resistance to serum is strain-dependent (multifactorial ANOVA, $p < 0.001$).

Second, we explored the ability of capsule mutants to survive in the presence of primary (sodium cholate –CHO-) and secondary (sodium deoxycholate –DCH-) bile salts, as they are antibacterial compounds that may disrupt bacterial cell membranes [34]. At physiological concentrations (0.05%), we observed no significant sensitivity of *Kpn* SC to bile salts

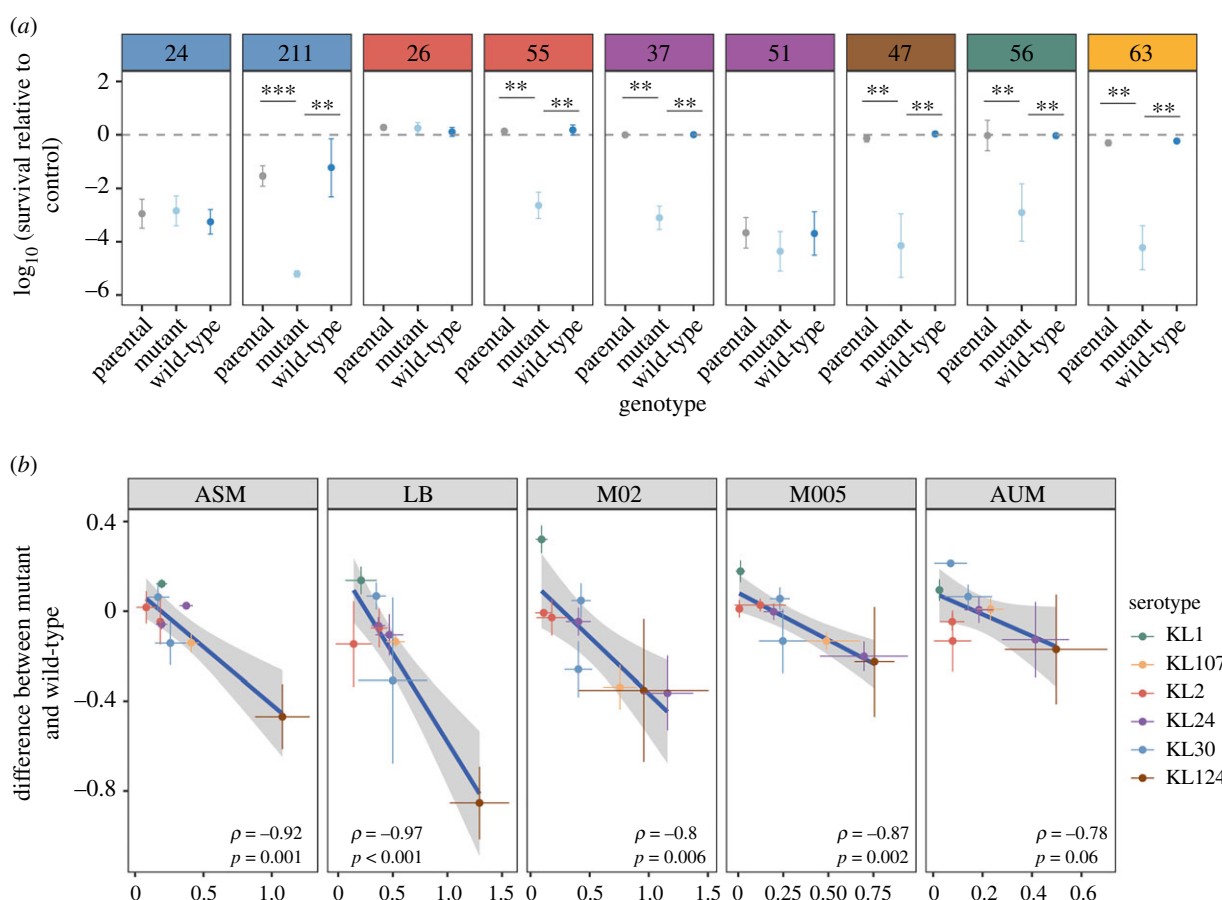

**Figure 4.** Fate of capsule mutants in host-related context. (*a*) Resistance to human serum. Data are presented as relative survival to control (heat-inactivated human serum). *$p < 0.05$; **$p < 0.01$; ***$p < 0.001$ paired *t*-test. Only significant statistics are represented. (*b*) Correlation between the total amount of biofilm formed by the wild-type and the difference in biofilm formation between the wild-type and the deletion mutant across environments. *P*-values correspond to Spearman's rank correlation. Results are qualitatively similar when only taking into account the five strains with no off-target mutations. Absolute values of biofilm formation are shown in the electronic supplementary material, figure S9. (Online version in colour.)

(electronic supplementary material, figure S8). When treated with 10X physiological concentrations, some, but not all, capsule mutants showed reduced viability (electronic supplementary material, figure S8).

Finally, the ability of niche colonization, especially within a host, is directly linked to the ability of the strains to adhere to surfaces and proliferate, that is, to form a biofilm. On average, there is no difference in the amount of biofilm between Cap+ and Cap− strains ($p > 0.05$, $n = 9$, paired *t*-test, in all different growth media). However, the amount of biofilm formation in the parental strain is negatively correlated with the difference in biofilm formation between the capsule mutant and its wild-type (figure 4*b*). In other words, the capsule tends to contribute positively to biofilm formation in strains forming more biofilm. At the strain level, we observe that some Cap− adhere significantly more across all environments (strain KL1.no. 56) (electronic supplementary material, figure S9), whereas other strains have a stronger ability to form biofilm when the capsule is present (strain KL124.no. 47) (electronic supplementary material, figure S9). This might depend on the capsule composition or it may also result from the absence of adhesion factors in the strain. Overall, our data suggest that it is not the presence or absence of the capsule that affects the ability to form biofilms, but rather the amount of capsule expressed by a given strain.

Taken together, these results show that some functions are not inherent to the presence of the capsule, but are rather strain-dependent, or dependent on the serotype-genome interaction.

## (f) Complex interactions between the environment, genetic background and serotype

To further understand the interactions between the serotype, genetic background and environment, we analysed different capsule-related variables in the different growth media. More precisely, we assessed capsule production (electronic supplementary material, figure S1), the natural emergence of non-capsulated mutants (figure 3), growth rate, yield and biofilm formation, and hypermucoviscosity of strains, a trait linked to hypervirulence and driven, at least partly by the amount of capsule [35] (electronic supplementary material, figures S10–S12), in all 19 strains used in the evolution experiment. First, we used principal components analysis (PCA) to analyse how the different factors (media, serotype or strain) explain the observed variance in the data. The first and second principal components explained more than 60% of the variation observed (figure 5*a*). We observed no clear grouping of serotypes according to the two first axes, suggesting that the serotype is not a major determinant of the capsule-related phenotypes (electronic supplementary material, figure S13). The ellipses regrouping the points by type of media show a small overlap between both rich media (ASM and LB) and these are clearly separated

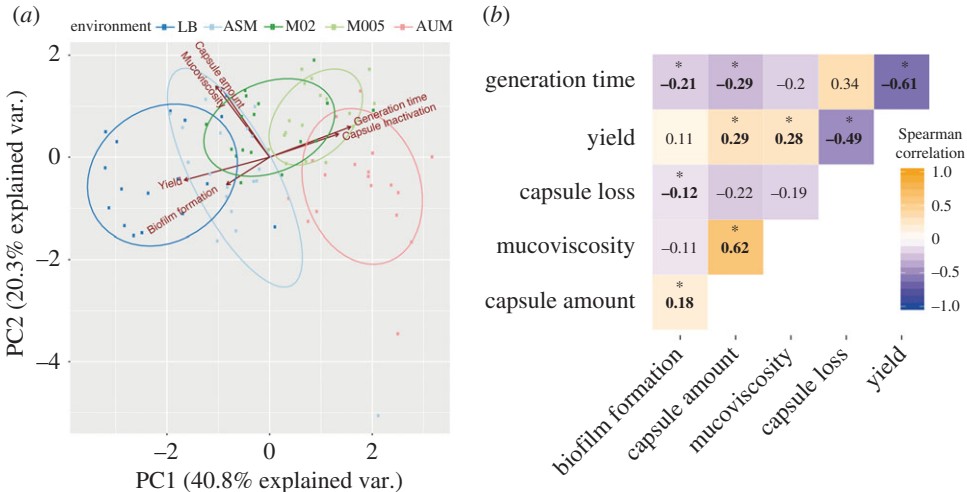

**Figure 5.** The environment, not the serotype, explains most of the variance observed across capsule-associated traits. (*a*) Principal component analyses (PCA) of capsule-related phenotypes. Phenotypes are coloured by environment. Each individual point represents a strain. PCA was performed using the *prcomp* function (options scale and centre = TRUE) in R and we used the ggbiplot package for visualization. (*b*) Correlation matrix of all traits. Numbers indicate Spearman's correlation. Asterisks identify significant correlations ($p < 0.05$). (Online version in colour.)

from the two poorest media (M005 and AUM). This suggests that most of the variance is explained by the media, presumably by the amount of nutrients available (figure 5*a*). We then used multifactorial ANOVA to analyse quantitatively the effect of the growth media and the serotype on each trait. As expected, we observed a significant effect of the growth media in all measured traits, except in biofilm formation ($R^2 = 0.57$, $p = 0.06$, electronic supplementary material, table S3). As suggested by the PCA, the influence of the serotype in the capsule-associated traits is less important. Interestingly, the terms of an interaction between the environment and the serotype are not statistically significant, suggesting that they are independent (electronic supplementary material, table S3).

We also sought to determine how the different capsule-related variables correlated with each other. The PCA analysis already suggested the existence of such correlations (figure 5*b*). As expected, we observe a strong correlation between hyper-mucoviscosity and the amount of capsule, in all individual growth media except in M005. Consistently with previous reports [36,37], we observe that the population yield is inversely proportional to the growth rate (figure 5*b*). This is significant in all growth media, except M005 (for which $p = 0.07$). Further, the amount of capsule is positively correlated with the maximum yield (figure 5*b*), supporting our observation that the capsule contributes to increased yield of the population. By contrast, there is a negative correlation between the amount of capsule and the generation time, a result that is driven by growth in nutrient-poor environments (figure 5*b*). More precisely, in poor media, bacteria with more capsule are able to grow faster that those producing less capsule. Overall, our results show that the capsule may impact several important cellular traits, like mucoviscosity and growth, but this is strongly dependent on the environment.

## 3. Discussion

The capsule is an extracellular structure commonly associated with the ability to resist biotic or abiotic aggressions, like the immune system, phages or desiccation [9,25,26]. In our previous work, we showed that capsules were more prevalent in free-living environmental bacteria than in pathogenic

bacteria. This suggested that the role of the capsule in virulence may be a by-product of adaptation outside the host, raising the question of its primary function in cells [3]. Here, we explored the conditions in which the capsule is advantageous in the absence of biotic or abiotic aggressions despite the cost of its production and whether their function is conserved. The evolution experiments and the direct competitions showed that capsule production provides a disadvantage during growth in rich medium and an advantage during growth in nutrient-poor environments. This advantage, *ca* 20%, was independent of the capsule mutants used, either the outer membrane exporter $\Delta wza$ or the initial glycosyltransferase mutants, $\Delta wcaJ$. The large variation in the costs and the differences observed across strains in important pathogenesis-associated traits, suggest that the role of the capsule is not evolutionarily conserved across strains.

To explain the results of our competition experiments, we examined the effect of the capsule on the growth dynamics of wild-type strains relative to their capsule mutants. In ASM, the media allowing the highest carrying capacity, we observed that the capsule is costly since mutants grow faster and achieve higher yields. In LB, the results are more complex. As in ASM, Cap− readily emerge in the evolution experiments, but there are no significant differences in terms of growth rate between $\Delta wza$ and the wild-types. This was also observed in *K. pneumoniae* KPPR1, a strain not included in our panel [38]. However, we do note an increased yield and a larger AUC in Cap− strains, indicative of the cost of the capsule in LB. Despite the absence of an increased growth rate, the higher fitness of $\Delta wza$ mutants in competition experiments could still be explained by longer exponential phase periods.

Opposite to what we observe in nutrient-rich environments, when resources are scarce, the capsule increases growth rate and populations reach higher yields and thus provides a competitive advantage. Further, we observed a negative correlation between the amount of capsule and generation time in nutrient-poor environments (figure 5*b*). Bacteria producing more capsule are able to grow faster in nutrient-poor environments. These results are somewhat counterintuitive, as in nutrient-poor media, one would

expect to observe preservation of sugars for essential functions and cell structures. Our attempts to understand how the capsule increases population yield or growth rate, either by assessing death rate, cell survival or the ability to consume the capsule did not provide a clear explanation. We speculate that the capsule plays an important role in cell homeostasis and its disruption may affect the local density and function of surface proteins, such as nutrient uptake systems, and a range of exported molecules important for cell growth, like siderophores. The nutrient-poor media AUM contains ten times less iron than the nutrient-rich LB [28]. In such an iron-depleted environment, iron-acquisition could be the limiting growth factor, and the capsule may play an important role in its uptake both at the physical and regulatory level. At the regulatory level, IscR, RhyB and Fur which control several iron-acquisition genes are also able to regulate directly and/or indirectly the capsule operon [39,40]. Inversely, it could also be that the levels of the capsule (or lack thereof) could affect the expression of siderophores, and thus, bacterial growth. At the physical level, the spatial structure generated by the capsule may act as a 'molecular sink' limiting siderophore dispersal, and accelerating iron uptake in the wild-type. By contrast, non-capsulated cells may facilitate free diffusion of siderophores resulting in less efficient iron uptake.

The capsule may also generate a gradient of nutrients from the environment to the outer membrane. Its absence would result in increased local concentrations of nutrients. It is already known that bacteria growing in nutrient-poor environments often have transporters that are sensitive to high-nutrient concentrations. If this is the case for *Kpn*, and nutrient uptake systems in poor media differ from those expressed in our rich media, capsule inactivation might result in rapid storage of the rare resource, which would delay cellular reproduction and growth [41]. Fast nutrient uptake could generate growth imbalances, the accumulation of toxic substances or in increase internal osmotic pressure [42], as observed when *Escherichia coli* cells pre-exposed to a lactose-poor environment are shifted to a lactose-rich environment [43].

Selection for capsule production in poor media challenges the current view that the primary role of the capsule is to withstand biotic aggressions. Our evaluation of capsule benefits on traits that might be advantageous during infection confirms this view. We observe that the capsule's role in resistance to human serum is not conserved across strains, as previously proposed [44]. Similarly, it was shown that in *K. pneumoniae* C105 (K35), capsule production masked the fimbriae required to attach to the host mucosa [45] and reduced biofilm formation. Here, by assessing a larger panel of strains, we reveal that this is not always true. The large variation in costs and roles highlighted here shows that the capsule role is not conserved across strains and suggests that its function may depend on other elements of the genome or on the serotype. For instance, some serotypes are associated with hypervirulence, like K1 and K2, whereas other serotypes are associated with multi-drug resistance like KL107 [46]. An experiment in which the capsule locus was exchanged between strains with a K2 and K21a serotype, suggested that the capacity to escape from macrophages was inherited with the capsule serotype [47]. Further, our experiments also contrast with previous reports in *S. pneumoniae*, in which it was suggested that capsulated

bacteria could use the capsules' polysaccharides to fuel faster growth and further multiply after consumption of all the environmental resources [31]. The fact that the roles of the capsule during pathogenesis are not conserved within and across species also supports the notion that the benefits of the capsule for virulence are a by-product of adaptation in non-host-related contexts. Overall, our analyses of multiple strains with different serotypes revealed that the role of the capsule in traits that may be relevant during pathogenesis is complex and depends not only on the strain but also the serotype.

To conclude, the complex relationship between the capsule and the host genome highlights that there are yet many research venues to be explored in terms of selection forces acting on the maintenance and diversification of the capsule, its function across a diversity of strains, and potential epistasy between the capsule and the rest of the genome. Here, we show that the growth media affects the relative fitness of capsulated bacteria and alone justifies the maintenance of the capsule in a population. Most literature suggests that the rapid turnover of capsular serotypes is owing to biotic pressures, such as phage predation, cell-to-cell interactions, the host adaptive immune system or protozoa grazing [48,49], but more work is needed to elucidate whether the abiotic conditions of the environment also play a role in the diversification of the capsule composition. For example, different capsules could be better adapted to certain poor environments because of their different propensity to aggregate or because of their organization at the cell surface [50,51]. It could also be that different serotypes are associated with specific functions or fitness advantages [47]. This might lead to the selection of serotype switching when bacteria colonize novel environments. Finally, more work is needed to address the multiple and distinct roles of capsules both within and across species [51]. Taken together, our work suggests that the primary evolutionary forces selecting for the existence of a capsule may be very different from those commonly evoked and highlights that much work is still needed to fully understand the primary role of capsules, both at the molecular and at the ecological level.

## 4. Material and methods

### (a) Bacterial strains and growth conditions
Bacterial strains are named as follows: KLXX.no. YY, where XX the capsule serotype, as determined by Kaptive [18], and YY the strain number. Bacteria and plasmids used in this study are listed in the electronic supplementary material, table S1. Kanamycin (50 µg ml$^{-1}$) or Streptomycin (100 µg ml$^{-1}$ for *E. coli* and 200 µg ml$^{-1}$ for *Klebsiella*) were used in the selection of the plasmids strain selection and cultures.

### (b) Growth media description
AUM and ASM were prepared as described previously [27,28]. AUM is mainly composed of 1% urea and 0.1% peptone with trace amounts of lactic acid, uric acid, creatinine and peptone. ASM is composed of 0.5% mucin, 0.4% DNA, 0.5% egg yolk and 0.2% aminoacids. LB is composed of 1% tryptone, 1% NaCl and 0.5% yeast extract. M02 and M005 indicate minimal M63B1 supplemented with 0.2% and 0.05% glucose (sole carbon source), respectively.

## (c) Natural inactivation of capsule

Three independent clones of 19 different strains were grown overnight in LB. To initiate the short evolution experiment, each culture was diluted at 1 : 100 in the different environments (ASM, LB, M02, M005, AUM) in 4 ml. Cultures were allowed to grow for 24 h at 37°C and diluted again to 1 : 100 in fresh media. This was repeated three times. Each culture in each environment was serially diluted and colony forming units (CFUs) were counted (three plates per sample, more than 100 colonies per plate). Results are expressed as a ratio of Cap− colonies observed over the total plate count. The limit of detection of Cap− in this assay was 0.22% ± 0.04. Cap− are easily visualized by the naked eye as mutants produce smaller, rough and translucent colonies. To ensure that the naturally emerged non-capsulated colonies observed on the plate are *bona fide* genetic mutants and not phase variants, we randomly chose 10 populations, using the *sample* function in R. Three non-capsulated clones from each population were randomly selected and restreaked twice. All clones derived from the original non-capsulated colony remained non-capsulated, demonstrating that the capsule inactivation is genetic. We also verified that the above-mentioned subset of clones was non-capsulated by the uronic acid method (electronic supplementary material, figure S2).

## (d) Generation of Δ*wza* and Δ*wcaJ* capsule mutants

Isogenic capsule mutants were constructed by an in-frame deletion of *wza* and *wcaJ* by allelic exchange. Upstream and downstream sequences of each gene (greater than 500 pb) were amplified using Phusion Master Mix (Thermo Scientific) then joined by overlap PCR. Primers are listed in the electronic supplementary material, table S4. The resulting PCR product was purified using the QIAquick Gel Extraction Kit and then cloned with the Zero Blunt® TOPO® PCR Cloning Kit (Invitrogen) into competent *E. coli* DH5α strain. KmR colonies were isolated and checked by PCR. Cloned Zero Blunt® TOPO® plasmid was extracted using the QIAprep Spin Miniprep Kit and digested for 2 h at 37°C with ApaI and SpeI restriction enzymes and ligated with T4 DNA ligase (Promega) overnight at 16°C to a double-digested pKNG101 plasmid. The ligation was transformed into competent *E. coli* DH5α-pir strain, and again into *E. coli* MFD λ-pir strain [52], used as a donor strain for conjugation into *Kpn* SC. Single cross-over mutants (transconjugants) were selected on Streptomycin plates and double cross-over mutants were selected on LB without salt and supplemented with 5% sucrose at room temperature. From each double-recombination, a mutant and a wild-type were isolated. Deletion mutants were verified by Illumina sequencing for off-target mutations by direct comparison to their respective reference genomes using *snippy* (v. 4.4.0 https://github.com/tseemann/snippy) and further verified with *breseq* (0.26.1 [53]) with default parameters.

## (e) Capsule extraction and quantification

The bacterial capsule was extracted as described in [54] and quantified by using the uronic acid method [55]. The uronic acid concentration in each sample was determined from a standard curve of glucuronic acid.

## (f) Competition assay

Mutants and their respective wild-types were grown overnight in LB. They were then mixed in a 1 : 1 proportion. A sample was taken and used for serial dilution and CFU counting as control of $T_0$. The co-culture was then diluted 1 : 100 in 4 ml of different environments (i.e. LB, AUM, ASM, M02 and M005). For control experiments, ASM and LB were diluted 1 : 10 in M63B1 with no supplement of glucose. After 24 h of competition ($T_{24}$), each mixture was serially diluted and plated. Capsulated and non-capsulated colonies are clearly differentiated visually and counted separately. The competitive index of capsulated strains was calculated by dividing the ratio of CFU at $T_{24}$ over $T_0$.

## (g) Growth curves

Overnight cultures were diluted at 1 : 100 in the different environments. Two hundred microliters of each subculture was transferred in a 96-well microplate. Absorbance ($OD_{600}$) of cell cultures was measured with a TECAN Genios™ plate reader. Absorbance values from within-block technical replicates were averaged and these averages were used as statistically independent data points. (i) Growth rate. Minimum generation times were estimated across replicates for the 1 h interval (ΔT) spanning the fastest growth during the exponential growth phase. This was calculated as follows:

$$\frac{1}{Td} = \frac{\log_{10}(\text{maximum}_{OD600}/\text{minimum}_{OD600})}{\log_{10}(2) * \Delta T}.$$

(ii) Maximum yield. Relative yields were calculated measuring ratios of the maximum absorbance of the assay compared to the reference (parental strain). Control experiments were performed to assess the correlation between CFU and $OD_{600}$ during growth for both capsulated (Spearman's ρ = 1.0, *p* = 0.002) and Cap− (Spearman's ρ = 0.94, *p* = 0.016). To confirm the differences in yield observed in the minimal medium between Cap+ and Cap− strains, we performed a control experiment (*n* = 3) in which we grew the strains KL1.no. 56 and KL30.no. 24 in micro-titerplates as described before. After 16 h, we plated an aliquot to count CFUs. Significantly less CFUs were observed in the Cap− compared to wild-type (*p* < 0.05 for both strains). (iii) AUC. AUC was calculated using the R function *trapz* from the *pracma* package.

## (h) Serum resistance

The ability of *Kpn* SC strains to resist killing by serum was performed as described previously [16,56,57]. Briefly, bacteria were grown in LB to OD 600 nm of 1, pelleted by centrifugation and resuspended in sterile phosphate buffered saline 1X. Two hundred microliters of bacterial suspension was added to 400 µl of pre-warmed at 37°C human sera (Sigma-Aldrich S7023), and the mixture was incubated for 2 h at 37°C. Control reactions were performed with serum inactivated by heat at 56°C for 30 min. Reactions were stopped by placing on ice, and viable bacterial counts were determined before and after incubation by serial dilution and CFU plating.

## (i) Other methods

See the electronic supplementary material, Methods.

Data accessibility. Data used in this study are available in Dryad Digital Repository: https://doi.org/10.5061/dryad.dz08kprwn [58].

Authors' contributions. O.R. and E.P.C.R. conceived, designed and coordinated the study. A.B. and O.R. carried out the experimental laboratory work and performed data analysis. O.R. performed the statistical analyses and wrote the first draft of the manuscript. E.P.C.R. critically revised the manuscript. All authors gave final approval for publication and agree to be held accountable for the work performed therein.

Competing interests. We declare we have no competing interests.

Funding. This work was supported by an ANR JCJC (Agence national de recherche) grant no. (ANR 18 CE12 0001 01 ENCAPSULATION) awarded to O.R. The laboratory is funded by a Laboratoire d'Excellence 'Integrative Biology of Emerging Infectious Diseases' grant (ANR-10-LABX-62-IBEID), the INCEPTION programme (PIA/ANR-16-CONV-0005) and the FRM (EQU201903007835). The funders had no role in the study design, data collection and interpretation, or the decision to submit the work for publication.

**Acknowledgements.** We thank Matthieu Haudiquet, Jean-Marc Ghigo and Nienke Buddelmeijer for critical reading of the manuscript. We thank Sylvain Brisse for providing us with the necessary *Klebsiella* strains, and Christiane Forestier and Damien Balestrino for providing the pKNG101 plasmid. We also thank Jean-Marc Ghigo and Christophe Beloin for the gift of *E. coli* S17 MFD λ-pir strain and *E. coli* DH5α-pir.

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
