## [Peer Review File · Proceedings of the Royal Society B: Biological Sciences]

Review History

RSPB-2020-2876.R0 (Original submission)

Review form: Reviewer 1

Recommendation

Major revision is needed (please make suggestions in comments)

Scientific importance: Is the manuscript an original and important contribution to its field?

Good

General interest: Is the paper of sufficient general interest?

Good

Quality of the paper: Is the overall quality of the paper suitable?

Good

Is the length of the paper justified?

Yes

Should the paper be seen by a specialist statistical reviewer?

Yes

Do you have any concerns about statistical analyses in this paper? If so, please specify them explicitly in your report.

No

It is a condition of publication that authors make their supporting data, code and materials available - either as supplementary material or hosted in an external repository. Please rate, if applicable, the supporting data on the following criteria.

Is it accessible?

N/A

Is it clear?

N/A

Is it adequate?

N/A

Do you have any ethical concerns with this paper?

No

Comments to the Author

This is an interesting paper showing that capsule production can be driven by nutrient condition in the environment and the results are counter-intuitive. This makes it potentially paradigm changing. The key finding is that consistent among different strains of *Klebsiella* and even within a few of the species, nutrient poor environment selects for capsule while nutrient rich environment exerts a small fitness cost.

The article is generally well organized and well written. There is only one major point that I would bring up and some minor points.

1. Major point: There is no mention of what kind of genetic mutations these non-capsulated mutants have. In fact, it is not clear that they do not have capsules except for the phenotype on the plates. Even though the authors later use deletion mutants in *wcaJ* and *wza* which recapitulate their initial fitness findings, it is important to select some of these naturally arisen mutants for sequencing.
2. This is important on 2 counts. First to see if inactivating mutations arise in the capsule locus and results in loss of capsule, which can be verified by SEM or uronic acid assay. Second, the capsule associated growth in nutrient poor condition and vice versa in nutrient rich condition may not necessarily means capsule drives growth in nutrient poor condition, as the authors claimed. The capsule loss could be associated with another unidentified trait such as exposure of membrane proteins involved in sensing nutrient condition (eg two component sensor regulator systems).
3. Minor point : have the authors examined iron acquisition systems in nutrient rich and poor conditions to see if upregulation of these systems could be associated with increased growth and whether capsule's presence affects iron acquisition rather than the other way round that is traditionally understood?
4. Minor point: Figure 2 is unclear as to what species mean in the legend.
5. Minor point: It is easier if authors name their strains with a prefix indicating serotype eg KL1xxxx (xxxx is strain number).
6. Minor point: Have the authors tried to determine capsule loss in sera besides the 5 media?

Review form: Reviewer 2

Recommendation

Accept with minor revision (please list in comments)

Scientific importance: Is the manuscript an original and important contribution to its field?

Good

General interest: Is the paper of sufficient general interest?

Good

Quality of the paper: Is the overall quality of the paper suitable?

Acceptable

Is the length of the paper justified?

Yes

Should the paper be seen by a specialist statistical reviewer?

Yes

Do you have any concerns about statistical analyses in this paper? If so, please specify them explicitly in your report.

No

It is a condition of publication that authors make their supporting data, code and materials available - either as supplementary material or hosted in an external repository. Please rate, if applicable, the supporting data on the following criteria.

Is it accessible?

Yes

Is it clear?

Yes

Is it adequate?

Yes

Do you have any ethical concerns with this paper?

No

Comments to the Author

In this study, Buffet et al uses a combination of experimental evolution and knock out experiments to examine capsule maintenance in various strain/serotype backgrounds and media in order to investigate the role of the capsule in processes such as growth and pathogenicity. This was an interesting study and my concerns raised below are of a relatively minor nature, as follows:

Line 31: please elaborate on the 'evolutionary challenges' i.e. survival? Competition?

Line 33: "capsules are encoded in half of the bacterial genomes" - the wording is awkward, suggest replaced genomes with species? Strains?

Lines 39-40: would be good to clarify whether the different combinations/residue modifications associate with different genes/alleles of the capsule synthesis locus, and whether this is the same in bacteria, archaea and yeast.

Line 41: it seems odd that the authors lead by mentioning serotype diversity being well studied in *S. pneumoniae* and *A. baumannii* and then don't elaborate further on these species but instead

immediately go on to talk about serotype diversity in *K. pneumoniae*.

Line 50: the authors mention here that they use *Klebsiella* as a model, but then go on to say in a few lines after that they only use strains belonging to the *K. pneumoniae* species complex.

Perhaps it would be more accurate to refer to the species complex instead of the whole genus at line 50 (and throughout the manuscript) as the *Klebsiella* genus as a whole is incredibly diverse and observations/results generated for the species complex strains in this study may not be applicable to the entire genus.

Line 69: unclear what "carrying capacities" means

Line 81: would be useful to state the K locus and ST of strain 62.

Line 81: "further, capsule maintenance was observed at least for 30 days in M02 in a KL2 strain" - the authors might need to elaborate on this as the relevance or significance of this result is unclear. Wasn't capsule maintenance also observed in other strains in minimal media?

Lines 88-89: "Wza is a core gene..." - the first part of this sentence refers to wza as a gene, in which case wza needs to be italicised, but the second part of the sentence refers to it as a protein product.

Line 101: typo, "mutants"

Lines 105-106: would be helpful to provide the K locus and maybe ST of these three strains. Here, it is unclear what exactly the authors mean by "competing" the strains - this would imply co-growing of multiple strains; did they co-grow the three strains together? If the strains were simply grown in diluted LB (to compare with the growth in non-diluted LB), then the term compete would not be appropriate.

Line 124-125: what about the growth rate, how did this compare to that observed in ASM?

Line 138: "... we observed no significant of the capsule" - wording needs to be corrected

Lines 135-149: This section of the results explores how the capsule confers increased growth in nutrient poor media and while the authors are able to disprove two hypotheses that may account for this, they don't actually resolve this i.e. we still don't know if or how the capsule accounts for increased growth in nutrient poor media. The authors highlight this in the discussion, but some commentary here in the results section about the "non-resolution" is also warranted.

Line 165: specially is a typo - especially?

Line 181-182: "to further understand the differences, we used the 19 strains" - what do the authors mean by "used", it would be informative to briefly outline the type of analyses performed in this section i.e. succinctly combine the first two sentences of this paragraph

Line 236: "... that the role capsule in resistance..." typo

Line 237: please specify which *Klebsiella* species

Line 268: "... were done as ..." do the authors mean to use "prepare" instead of "done"?

Lines 269-271: it would be useful to provide some meaningful measure of the composition (i.e. exact amounts of % composition)

Line 272: this reference isn't correct - please check that all references throughout the manuscript are correct.

Figures:

Fig 3, Fig S5, Fig S6A: the colours don't stand out as much against the blues of the bar plots - might be better to use shades of grey for the bar plots instead.

Fig S1: should the y axis be labelled with total amount of uronic acid instead of capsule?

Fig S2: no visual legend provided; unclear what the different shades of points represent. The shades of grey are also difficult to differentiate - perhaps using colour would be better.

Fig S3: the grey points against the grey and darker blue backgrounds of the bar plots are difficult to visualise.

Fig S7: it would be useful to also include a definition for 'CHO' and 'DCH' in the figure legend

Fig S9: re-order either the title or panels A and B of the figure to match the order presented within the figure. What is the unit of measurement for the generation time (minutes?)

Fig S12: difficult to see the points - increase the size and/or shade

Tables:

Table S2: gene names need to be italicised. What are the roles of *pcaR* and *sacX*?

Decision letter (RSPB-2020-2876.R0)

09-Jan-2021

Dear Dr Rendueles:

Your manuscript has now been peer reviewed and the reviews have been assessed by an Associate Editor. The reviewers' comments (not including confidential comments to the Editor) and the comments from the Associate Editor are included at the end of this email for your reference. As you will see, the reviewers and the Editors have raised some concerns with your manuscript and we would like to invite you to revise your manuscript to address them.

Research ethics:

Use of animals and field studies:

It is a condition of publication that you make available the data and research materials supporting the results in the article. Please see our Data Sharing Policies (<https://royalsociety.org/journals/authors/author-guidelines/#data>). Datasets should be deposited in an appropriate publicly available repository and details of the associated accession number, link or DOI to the datasets must be included in the Data Accessibility section of the

article (<https://royalsociety.org/journals/ethics-policies/data-sharing-mining/>). Reference(s) to datasets should also be included in the reference list of the article with DOIs (where available).

Please submit a copy of your revised paper within three weeks. If we do not hear from you within this time your manuscript will be rejected. If you are unable to meet this deadline please let us know as soon as possible, as we may be able to grant a short extension.

Best wishes,
Dr Sasha Dall
mailto: proceedingsb@royalsociety.org

Associate Editor

Board Member: 1

Comments to Author:

In this study, the authors report a range of new results on the benefits and costs of capsules in *Klebsiella*, showing that capsules are readily lost in nutrient-rich media but, counterintuitively, are maintained and seem to be beneficial in nutrient-poor media. This challenges the paradigm of capsules being primarily an adaptation to pathogen survival within their hosts. I agree with the two reviewers that this is a very interesting article. It addresses the fundamental but elusive question of why many bacteria produce capsules and hence should be of interest to many biologists.

Reviewer 1 raises an important point about the lack of information about the genetic basis of capsule loss in the evolution experiments. I agree that it would be good to perform whole-genome sequencing on some of the Cap- mutants to see in what genes the mutations occurred.

Given that the authors did Illumina sequencing to confirm their deletion mutants, it appears that they have the technology and bioinformatics pipelines in place to do this.

One point which I think should be clarified is the evolution experiment at the beginning of the paper. In the Methods section, it sounds as if there was only a single evolving population for each strain and environment. However, in the Results section the authors mention a total of 290 independently evolved populations, which indicates roughly three ($290/(5 \times 19)$) replicate populations? This would be in line with the caption in Fig. 1, but why were sometimes more than three replicate populations propagated, and not always exactly three? Also, if the data in Fig. 1 is just an average of three or four data points, I think instead of showing the SD it would be better to show the actual frequencies for each population as small points.

Another minor comment: like the authors and reviewer 2, I think the result that the capsule is beneficial in nutrient-poor environments is a rather counter-intuitive because capsule production obviously requires nutrients. There is a short paragraph about this in the Discussion but I think it would be good to elaborate on this and perhaps come up with some hypotheses for how the capsule could be beneficial in such environments. This would of course have to be speculative, but it might stimulate others to test some of the ideas put forward.

Reviewer(s)' Comments to Author:

Referee: 1

Comments to the Author(s)

This is an interesting paper showing that capsule production can be driven by nutrient condition in the environment and the results are counter-intuitive. This makes it potentially paradigm changing. The key finding is that consistent among different strains of *Klebsiella* and even within a few of the species, nutrient poor environment selects for capsule while nutrient rich environment exerts a small fitness cost.

The article is generally well organized and well written. There is only one major point that I would bring up and some minor points.

1. Major point: There is no mention of what kind of genetic mutations these non-capsulated mutants have. In fact, it is not clear that they do not have capsules except for the phenotype on the plates. Even though the authors later use deletion mutants in *wcaJ* and *wza* which recapitulate their initial fitness findings, it is important to select some of these naturally arisen mutants for sequencing.
2. This is important on 2 counts. First to see if inactivating mutations arise in the capsule locus and results in loss of capsule, which can be verified by SEM or uronic acid assay. Second, the capsule associated growth in nutrient poor condition and vice versa in nutrient rich condition may not necessarily means capsule drives growth in nutrient poor condition, as the authors claimed. The capsule loss could be associated with another unidentified trait such as exposure of membrane proteins involved in sensing nutrient condition (eg two component sensor regulator systems).
3. Minor point : have the authors examined iron acquisition systems in nutrient rich and poor conditions to see if upregulation of these systems could be associated with increased growth and whether capsule's presence affects iron acquisition rather than the other way round that is traditionally understood?
4. Minor point: Figure 2 is unclear as to what species mean in the legend.
5. Minor point: It is easier if authors name their strains with a prefix indicating serotype eg KL1xxxx (xxxx is strain number).
6. Minor point: Have the authors tried to determine capsule loss in sera besides the 5 media?

Referee: 2

Comments to the Author(s)

In this study, Buffet et al uses a combination of experimental evolution and knock out experiments to examine capsule maintenance in various strain/serotype backgrounds and media

in order to investigate the role of the capsule in processes such as growth and pathogenicity. This was an interesting study and my concerns raised below are of a relatively minor nature, as follows:

Line 31: please elaborate on the 'evolutionary challenges' i.e. survival? Competition?

Line 33: "capsules are encoded in half of the bacterial genomes" - the wording is awkward, suggest replaced genomes with species? Strains?

Lines 39-40: would be good to clarify whether the different combinations/residue modifications associate with different genes/alleles of the capsule synthesis locus, and whether this is the same in bacteria, archaea and yeast.

Line 41: it seems odd that the authors lead by mentioning serotype diversity being well studied in *S. pneumoniae* and *A. baumannii* and then don't elaborate further on these species but instead immediately go on to talk about serotype diversity in *K. pneumoniae*.

Line 50: the authors mention here that they use *Klebsiella* as a model, but then go on to say in a few lines after that they only use strains belonging to the *K. pneumoniae* species complex.

Perhaps it would be more accurate to refer to the species complex instead of the whole genus at line 50 (and throughout the manuscript) as the *Klebsiella* genus as a whole is incredibly diverse and observations/results generated for the species complex strains in this study may not be applicable to the entire genus.

Line 69: unclear what "carrying capacities" means

Line 81: would be useful to state the K locus and ST of strain 62.

Line 81: "further, capsule maintenance was observed at least for 30 days in M02 in a KL2 strain" - the authors might need to elaborate on this as the relevance or significance of this result is unclear. Wasn't capsule maintenance also observed in other strains in minimal media?

Lines 88-89: "Wza is a core gene..." - the first part of this sentence refers to *wza* as a gene, in which case *wza* needs to be italicised, but the second part of the sentence refers to it as a protein product.

Line 101: typo, "mutants"

Lines 105-106: would be helpful to provide the K locus and maybe ST of these three strains. Here, it is unclear what exactly the authors mean by "competing" the strains - this would imply co-growing of multiple strains; did they co-grow the three strains together? If the strains were simply grown in diluted LB (to compare with the growth in non-diluted LB), then the term compete would not be appropriate.

Line 124-125: what about the growth rate, how did this compare to that observed in ASM?

Line 138: "... we observed no significant of the capsule" - wording needs to be corrected

Lines 135-149: This section of the results explores how the capsule confers increased growth in nutrient poor media and while the authors are able to disprove two hypotheses that may account for this, they don't actually resolve this i.e. we still don't know if or how the capsule accounts for increased growth in nutrient poor media. The authors highlight this in the discussion, but some commentary here in the results section about the "non-resolution" is also warranted.

Line 165: specially is a typo - especially?

Line 181-182: "to further understand the differences, we used the 19 strains" - what do the authors mean by "used", it would be informative to briefly outline the type of analyses performed in this section i.e. succinctly combine the first two sentences of this paragraph

Line 236: "... that the role capsule in resistance..." typo

Line 237: please specify which *Klebsiella* species

Line 268: "... were done as ..." do the authors mean to use "prepare" instead of "done"?

Lines 269-271: it would be useful to provide some meaningful measure of the composition (i.e. exact amounts of % composition)

Line 272: this reference isn't correct - please check that all references throughout the manuscript are correct.

Figures:

Fig 3, Fig S5, Fig S6A: the colours don't stand out as much against the blues of the bar plots - might be better to use shades of grey for the bar plots instead.

Fig S1: should the y axis be labelled with total amount of uronic acid instead of capsule?

Fig S2: no visual legend provided; unclear what the different shades of points represent. The shades of grey are also difficult to differentiate - perhaps using colour would be better.

Fig S3: the grey points against the grey and darker blue backgrounds of the bar plots are difficult to visualise.

Fig S7: it would be useful to also include a definition for 'CHO' and 'DCH' in the figure legend

Fig S9: re-order either the title or panels A and B of the figure to match the order presented within the figure. What is the unit of measurement for the generation time (minutes?)

Fig S12: difficult to see the points - increase the size and/or shade

Tables:

Table S2: gene names need to be italicised. What are the roles of *pcaR* and *sacX*?

Author's Response to Decision Letter for (RSPB-2020-2876.R0)

See Appendix A.

RSPB-2020-2876.R1 (Revision)

Review form: Reviewer 1

Recommendation

Accept as is

Scientific importance: Is the manuscript an original and important contribution to its field?

Good

General interest: Is the paper of sufficient general interest?

Good

Quality of the paper: Is the overall quality of the paper suitable?

Good

Is the length of the paper justified?

Yes

Should the paper be seen by a specialist statistical reviewer?

No

Do you have any concerns about statistical analyses in this paper? If so, please specify them explicitly in your report.

No

It is a condition of publication that authors make their supporting data, code and materials available - either as supplementary material or hosted in an external repository. Please rate, if applicable, the supporting data on the following criteria.

Is it accessible?

N/A

Is it clear?

N/A

Is it adequate?

N/A

Do you have any ethical concerns with this paper?

No

Comments to the Author

This revision has addressed my concerns. There are minor grammatical errors to be taken care of during proofs. The discussion has been greatly improved and a pleasure to read.

Decision letter (RSPB-2020-2876.R1)

05-Feb-2021

Dear Dr Rendueles

I am pleased to inform you that your manuscript entitled "Nutrient conditions are primary drivers of bacterial capsule maintenance in *Klebsiella*" has been accepted for publication in Proceedings B.

Open Access

You are invited to opt for Open Access, making your freely available to all as soon as it is ready for publication under a CC BY licence. Our article processing charge for Open Access is £1700. Corresponding authors from member institutions (<http://royalsocietypublishing.org/site/librarians/allmembers.xhtml>) receive a 25% discount to these charges. For more information please visit <http://royalsocietypublishing.org/open-access>.

Paper charges

Sincerely,
Dr Sasha Dall
Editor, Proceedings B
mailto: proceedingsb@royalsociety.org

Associate Editor:

Board Member: 1

Comments to Author:

The authors have done an excellent job addressing the reviewers' and my own comments. In particular, I'm happy to see that the authors have been able to perform additional assays that confirm the absence of capsules in some of their mutants. I also think that the preprint ms will be a useful complement article to the present paper, so with that paper in place I don't think performing additional WGS of their mutants is essential. However, I noticed that the DOI number in the cited preprint on biorxiv does not seem to match the one in their cover letter; instead searching for it produced a different paper by the authors about prophages. Also, the title of the paper in the references list is different from the one that shows up when following the link in the cover letter. This needs to be corrected in the final version before publication.

Appendix A

Olaya RENDUELES

Microbial Evolutionary Genomics Unit
Department of Genomes & Genetics
INSTITUT PASTEUR
28 rue du docteur Roux
75015 Paris France

olaya.rendueles-garcia@pasteur.fr

19th January 2021

Dear Dr. Dall,

Thank you for the invitation to revise our manuscript in consideration of the reviewer's comments. We address them below, in each case explaining how we have revised the manuscript in light of the comment.

As requested, we have now better explained our evolution experiment in the methods section and provide a clearer Figure 1. We further confirmed that the spontaneous mutants are *bona fide* non-capsulated, by performing multiple capsule quantification assays. We believe that these assays alone are enough and do not require the report of the mutations accumulated in these clones to ensure that they are non-capsulated. Further, the focus of our work, as the editor and the reviewers pointed out, is to try to further understand in which environments the capsule constitutes an advantage and to ultimately understand if it evolved primarily as an adaptation to a host, during pathogenesis. In this context, we believe that including the mutations that lead to the capsule inactivation would drift away from the main message, and would not provide further answers to the questions raised.

We have nevertheless sequenced some non-capsulated clones that emerged during our experiments. We used these clones as an experimental confirmation of a much larger bioinformatics study in which we analyzed the capsule locus of over 4000 *Klebsiella* genomes. We believe those results are a better fit in that study in which we specifically address the genetic basis of the evolution of the capsule locus and how that affects the evolution of the species. That work can be consulted in the following preprint: <https://doi.org/10.1101/2020.12.09.417816>, which was not yet available at the time of our initial submission, and is now appropriately referenced in our manuscript and available to the community.

We would also like to highlight that we have changed some wording throughout the manuscript for clarity purposes and consistency with other works. We now refer to capsule inactivation instead of the loss of capsule.

We have also extended our discussion and speculate about potential explanations to the increased growth rate of capsulated strains in nutrient poor environments.

We hope you will find that we have effectively addressed the comments of the reviewers and your own, and that our improved manuscript is now suitable for publication in *Proceedings of the Royal Society*.

Best regards,

Olaya Rendueles

Associate Editor

Board Member: 1

Comments to Author:

In this study, the authors report a range of new results on the benefits and costs of capsules in *Klebsiella*, showing that capsules are readily lost in nutrient-rich media but, counterintuitively, are maintained and seem to be beneficial in nutrient-poor media. This challenges the paradigm of capsules being primarily an adaptation to pathogen survival within their hosts. I agree with the two reviewers that this is a very interesting article. It addresses the fundamental but elusive question of why many bacteria produce capsules and hence should be of interest to many biologists.

We thank the editor and the reviewers for the assessment of the importance of our work.

Reviewer 1 raises an important point about the lack of information about the genetic basis of capsule loss in the evolution experiments. I agree that it would be good to perform whole-genome sequencing on some of the Cap- mutants to see in what genes the mutations occurred. Given that the authors did Illumina sequencing to confirm their deletion mutants, it appears that they have the technology and bioinformatics pipelines in place to do this.

As we detailed in the letter and we further explain after Reviewer's 1 comment, we believe that reporting the mutations leading to non-capsulated cells do not provide further insights to the question we address here, namely, whether the capsule evolved primarily for adaptation to the host. In fact, we believe it will distract the focus of the manuscript. However, we agree that this provides relevant confirmation that the capsule locus is inactive. Hence, we now reference a preprint, <https://doi.org/10.1101/2020.12.09.417816> (that was not yet available at the time this manuscript was first submitted), in which we provide the WGS analyses of 22 non-capsulated clones from independent replicate populations stemming from 8 different ancestors. This analysis can be consulted in the context of a much larger study precisely addressing how the capsule operon evolves and diversifies, and, in term, affects the evolution of the species. As expected, this analysis revealed inactivating mutations in key genes of the capsule locus, thereby further demonstrating the loss of the capsule.

One point which I think should be clarified is the evolution experiment at the beginning of the paper. In the Methods section, it sounds as if there was only a single evolving population for each strain and environment. However, in the Results section the authors mention a total of 290 independently evolved populations, which indicates roughly three ($290/(5 \times 19)$) replicate populations? This would be in line with the caption in Fig. 1, but why were sometimes more than three replicate populations propagated, and not always exactly three?

We have clarified our methods section. Indeed, multiple independent clones of each strain evolved in parallel. For all strains, 3 replicate populations were propagated, except for KL2.#26 and KL1.#209, for which 4 clones were propagated in all environments.

Also, if the data in Fig. 1 is just an average of three or four data points, I think instead of showing the SD it would be better to show the actual frequencies for each population as small points.

We have modified Figure 1 and its legend to show each independent population instead of the mean and SD of three independent replicates.

Another minor comment: like the authors and reviewer 2, I think the result that the capsule is beneficial in nutrient-poor environments is a rather counter-intuitive because capsule production obviously requires nutrients. There is a short paragraph about this in the Discussion but I think it would be good to elaborate on this and perhaps come up with some hypotheses for how the capsule could be beneficial in such environments. This would of course have to be speculative, but it might stimulate others to test some of the ideas put forward.

We have now expanded our discussion to propose new hypotheses on the benefits of the capsule in nutrient poor environments.

Reviewer(s)' Comments to Author:

Referee: 1

Comments to the Author(s)

This is an interesting paper showing that capsule production can be driven by nutrient condition in the environment and the results are counter-intuitive. This makes it potentially paradigm changing. The key finding is that consistent among different strains of *Klebsiella* and even within a few of the species, nutrient poor environment selects for capsule while nutrient rich environment exerts a small fitness cost.

The article is generally well organized and well written. There is only one major point that I would bring up and some minor points.

We thank the reviewer for his comments and the time spent evaluating our work.

1.1. Major point: There is no mention of what kind of genetic mutations these non-capsulated mutants have. In fact, it is not clear that they do not have capsules except for the phenotype on the plates. Even though the authors later use deletion mutants in *wcaJ* and *wza* which recapitulate their initial fitness findings, it is important to select some of these naturally arisen mutants for sequencing.

The reviewer suggests we should analyze the sequences of the non-capsulated clones as a confirmation that these clones are non-capsulated. It would be impossible, under the current epidemic circumstances to perform the sequencing in the short period given for revision. Fortunately, as abovementioned, we already analyzed 22 non-capsulated clones (each from an independent replicate population) and stemming from 8 different ancestors as part of a study focusing precisely on the genetic basis of capsule inactivation and evolution. This manuscript has a preprint (<https://doi.org/10.1101/2020.12.09.417816>) and confirms that the mutants have accumulated mutations in the capsule operon.

To further confirm that these mutants are non-capsulated, we performed capsule quantification assays by the uronic acid method in three clones from 10 different and randomly chosen populations. This unequivocally shows that the spontaneous capsule mutants and that these clones are *bona fide* non-capsulated.

1.2. This is important on 2 counts. First to see if inactivating mutations arise in the capsule locus and results in loss of capsule, which can be verified by SEM or uronic acid assay.

As mentioned above, we have performed capsule quantification assays in 30 clones and show that they are *bona fide* non-capsulated.

Second, the capsule associated growth in nutrient poor condition and vice versa in nutrient rich condition may not necessarily mean capsule drives growth in nutrient poor condition, as the authors claimed. The capsule loss could be associated with another unidentified trait such as exposure of membrane proteins involved in sensing nutrient condition (eg two component sensor regulator systems).

The reviewer is right. Capsule inactivation may interfere with other membrane-associated proteins and alter their efficiency and thus cellular growth. However, the literature on this is very skim and these hypotheses remain speculative. We have expanded our discussion to include this, as well as other possibilities.

1.3. Minor point : have the authors examined iron acquisition systems in nutrient rich and poor conditions to see if upregulation of these systems could be associated with increased growth and whether capsule's presence affects iron acquisition rather than the other way round that is traditionally understood?

Iron is an important element for bacterial growth, especially in the host. Unfortunately, it is unclear its relative importance in the five different environments we use in this study.

In line with the previous comment, all molecules exported from the cell or that are located at its surface can be potentially influenced by the presence or absence of the capsule. Following the reviewer's comment, we modified our discussion to include the interference between capsule and iron.

1.4. Minor point: Figure 2 is unclear as to what species mean in the legend.

This has been reformulated.

1.5. Minor point: It is easier if authors name their strains with a prefix indicating serotype eg KL1xxxx (xxxx is strain number).

We have now changed the name of the strains throughout the manuscript text as suggested by the reviewer.

1.6. Minor point: Have the authors tried to determine capsule loss in sera besides the 5 media?

The results we present in this study show that some strains are readily killed by serum (3 out of 9 tested), independent of the presence/absence of capsule, which implies that they would not be able to grow and evolve in such an environment. We also show that in most non-capsulated mutants, human serum is able to efficiently kill cells. We thus predict that, given the large fitness disadvantage of non-capsulated mutants in this environment, even if they emerged, they would be very rapidly cleared and would not be observed. We believe evolving strains in serum to assess capsule inactivation will only be meaningful in the one strain in which capsule mutants would not have a large fitness disadvantage and in which growth can be sustained. We thus believe that performing this experiment will not be very informative in understanding capsule maintenance as it will most likely select for resistant mutants to sera.

Referee: 2

Comments to the Author(s)

In this study, Buffet et al uses a combination of experimental evolution and knock out experiments to examine capsule maintenance in various strain/serotype backgrounds and media in order to investigate the role of the capsule in processes such as growth and pathogenicity.

We thank the reviewer for carefully reading the manuscript.

This was an interesting study and my concerns raised below are of a relatively minor nature, as follows:

2.1 Line 31: please elaborate on the 'evolutionary challenges' i.e. survival? Competition?

We have now specified on the evolutionary challenges.

2.2 Line 33: "capsules are encoded in half of the bacterial genomes" - the wording is awkward, suggest replaced genomes with species? Strains?

These findings correspond to a bioinformatic search of all available genomes on NCBI. For further clarity, we added the notion of "sequenced genomes" in the text.

2.3 Lines 39-40: would be good to clarify whether the different combinations/residue modifications associate with different genes/alleles of the capsule synthesis locus, and whether this is the same in bacteria, archaea and yeast.

This has been modified in the main text.

2.4 Line 41: it seems odd that the authors lead by mentioning serotype diversity being well studied in *S. pneumoniae* and *A. baumannii* and then don't elaborate further on these species but instead immediately go on to talk about serotype diversity in *K. pneumoniae*.

We have now reformulated our sentence.

2.5 Line 50: the authors mention here that they use *Klebsiella* as a model, but then go on to say in a few lines after that they only use strains belonging to the *K. pneumoniae* species complex. Perhaps it would be more accurate to refer to the species complex instead of the whole genus at line 50 (and throughout the manuscript) as the *Klebsiella* genus as a whole is incredibly diverse and observations/results generated for the species complex strains in this study may not be applicable to the entire genus.

The reviewer is correct. We now exclusively refer to the *K. pneumoniae* species complex, using the acronym *Kpn SC* for simplicity.

2.6 Line 69: unclear what "carrying capacities" means

Carrying capacity refers to the maximum population size an environment can sustain and depends on the resources available. This is now clarified in the text.

2.7 Line 81: would be useful to state the K locus and ST of strain 62.

This is now mentioned in the text.

2.8 Line 81: "further, capsule maintenance was observed at least for 30 days in M02 in a KL2 strain" - the authors might need to elaborate on this as the relevance or significance of this result is unclear. Wasn't capsule maintenance also observed in other strains in minimal media?

Capsule is maintained for three days in all strains as shown in Figure 1. Here we tested whether it could be lost later on, or maintained for a long period of time. We thus allowed the evolution experiment to go on for 30 days.

The logic for this experiment is now better explained in the text.

2.9 Lines 88-89: "Wza is a core gene..." - the first part of this sentence refers to *wza* as a gene, in which case *wza* needs to be italicised, but the second part of the sentence refers to it as a protein product.

This has been corrected.

2.10 Line 101: typo, "mutants"

This has been modified.

2.11 Lines 105-106: would be helpful to provide the K locus and maybe ST of these three strains.

As suggested above, we have adopted the nomenclature *KX.#strain*, in which X is the serotype.

Here, it is unclear what exactly the authors mean by "competing" the strains - this would imply co-growing of multiple strains; did they co-grow the three strains together? If the strains were simply grown in diluted LB (to compare with the growth in non-diluted LB), then the term compete would not be appropriate.

We competed each strain against its respective capsule mutant in diluted LB.

This has been rephrased in the text.

2.12 Line 124-125: what about the growth rate, how did this compare to that observed in ASM?
This is specified in the previous lines, growth rate in ASM is also slower in capsulated strains.

2.13 Line 138: "... we observed no significant of the capsule" - wording needs to be corrected
This has been corrected.

2.14 Lines 135-149: This section of the results explores how the capsule confers increased growth in nutrient poor media and while the authors are able to disprove two hypotheses that may account for this, they don't actually resolve this i.e. we still don't know if or how the capsule accounts for increased growth in nutrient poor media. The authors highlight this in the discussion, but some commentary here in the results section about the "non-resolution" is also warranted.
This has been included at the end of the corresponding results section.

2.15 Line 165: specially is a typo - especially?
This has been corrected.

2.16 Line 181-182: "to further understand the differences, we used the 19 strains" - what do the authors mean by "used", it would be informative to briefly outline the type of analyses performed in this section i.e. succinctly combine the first two sentences of this paragraph
We have now changed the wording to avoid confusion and combined the two sentences as suggested by the reviewer.

2.17 Line 236: "... that the role capsule in resistance..." typo
This has been corrected.

2.18 Line 237: please specify which *Klebsiella* species
This reference corresponds to *K. pneumoniae* C105 (K35). It has been now specified in the text.

2.19 Line 268: "... were done as ..." do the authors mean to use "prepare" instead of "done"?
Prepare. This has been appropriately modified.

2.20 Lines 269-271: it would be useful to provide some meaningful measure of the composition (i.e. exact amounts of % composition)
We have now included the composition percentage of each growth medium.

2.21 Line 272: this reference isn't correct - please check that all references throughout the manuscript are correct.
We thank the reviewer for pointing this out. We have now thoroughly checked the references.

Figures:

2.22 Fig 3, Fig S5, Fig S6A: the colours don't stand out as much against the blues of the bar plots - might be better to use shades of grey for the bar plots instead.
The figures have been modified as suggested by the reviewer.

2.23 Fig S1: should the y axis be labelled with total amount of uronic acid instead of capsule?
This has been modified for accuracy purposes.

2.24 Fig S2: no visual legend provided; unclear what the different shades of points represent. The shades of grey are also difficult to differentiate - perhaps using colour would be better.
The different lines represent independently evolving populations. This is now clarified in the legend, and the colours modified for clarity.

2.25 Fig S3: the grey points against the grey and darker blue backgrounds of the bar plots are difficult to visualise.

The figure has been modified as suggested by the reviewer.

2.26 Fig S7: it would be useful to also include a definition for 'CHO' and 'DCH' in the figure legend

The legend now includes the precise definition of CHO and DCH.

2.27 Fig S9: re-order either the title or panels A and B of the figure to match the order presented within the figure. What is the unit of measurement for the generation time (minutes?)

Generation time is expressed in minutes. This is now specified in the y-axis label. The figure legend was reordered to match the panels.

2.28 Fig S12: difficult to see the points - increase the size and/or shade

The figure has been modified.

Tables:

2.29 Table S2: gene names need to be italicised. What are the roles of *pcaR* and *sacX*?

Genes have been italicized, and more details on gene functions has been provided.